# Dynamic Expression of Imprinted Genes in the Developing and Postnatal Pituitary Gland

**DOI:** 10.3390/genes12040509

**Published:** 2021-03-30

**Authors:** Valeria Scagliotti, Ruben Esse, Thea L. Willis, Mark Howard, Isabella Carrus, Emily Lodge, Cynthia L. Andoniadou, Marika Charalambous

**Affiliations:** 1Department of Medical and Molecular Genetics, Faculty of Life Sciences and Medicine, King’s College London, London SE19RT, UK; valeria.scagliotti@kcl.ac.uk (V.S.); ruben.esse@kcl.ac.uk (R.C.F.E.); isabellacarrus96@gmail.com (I.C.); 2Centre for Craniofacial and Regenerative Biology, Faculty of Dentistry, Oral and Craniofacial Sciences, King’s College London, London SE19RT, UK; thea.willis@kcl.ac.uk (T.L.W.); emily.lodge@kcl.ac.uk (E.L.); cynthia.andoniadou@kcl.ac.uk (C.L.A.); 3MRC Centre for Transplantation, Peter Gorer Department of Immunobiology, School of Immunology & Microbial Sciences, King’s College London, London SE19RT, UK; mark.howard@kcl.ac.uk; 4Department of Medicine III, University Hospital Carl Gustav Carus, Technische Universität Dresden, 01307 Dresden, Germany

**Keywords:** genomic imprinting, endocrinology, pituitary, growth, imprinting disorders, puberty, maternal behaviour, imprinted gene network, stem cells, pregnancy

## Abstract

In mammals, imprinted genes regulate many critical endocrine processes such as growth, the onset of puberty and maternal reproductive behaviour. Human imprinting disorders (IDs) are caused by genetic and epigenetic mechanisms that alter the expression dosage of imprinted genes. Due to improvements in diagnosis, increasing numbers of patients with IDs are now identified and monitored across their lifetimes. Seminal work has revealed that IDs have a strong endocrine component, yet the contribution of imprinted gene products in the development and function of the hypothalamo-pituitary axis are not well defined. Postnatal endocrine processes are dependent upon the production of hormones from the pituitary gland. While the actions of a few imprinted genes in pituitary development and function have been described, to date there has been no attempt to link the expression of these genes as a class to the formation and function of this essential organ. This is important because IDs show considerable overlap, and imprinted genes are known to define a transcriptional network related to organ growth. This knowledge deficit is partly due to technical difficulties in obtaining useful transcriptomic data from the pituitary gland, namely, its small size during development and cellular complexity in maturity. Here we utilise high-sensitivity RNA sequencing at the embryonic stages, and single-cell RNA sequencing data to describe the imprinted transcriptome of the pituitary gland. In concert, we provide a comprehensive literature review of the current knowledge of the role of imprinted genes in pituitary hormonal pathways and how these relate to IDs. We present new data that implicate imprinted gene networks in the development of the gland and in the stem cell compartment. Furthermore, we suggest novel roles for individual imprinted genes in the aetiology of IDs. Finally, we describe the dynamic regulation of imprinted genes in the pituitary gland of the pregnant mother, with implications for the regulation of maternal metabolic adaptations to pregnancy.

## 1. Introduction

Genomic imprinting is an epigenetically regulated process in mammals that results in the expression of autosomal genes from a single parental chromosome. These genes are commonly clustered in the genome into loci containing multiple imprinted genes (IGs). Within an imprinting locus, the allele-specific expression of both maternally and paternally expressed genes is commonly controlled by a regulatory element known as an imprinting centre (IC) or imprinting control region (ICR). ICRs display differential DNA methylation dependent upon their chromosomal origin; the epigenetic mark is acquired during germ cell development and maintained throughout the epigenetic reprogramming of the early embryo. Deletions of ICRs or epimutations that cause changes to the normal parental methylation pattern result in a locus-wide disruption to the IG expression dosage (recently reviewed in Tucci et al., 2019 [1]). Regulating the dosage of this important class of genes is crucial for the completion of development, since their products have essential roles in placentation and intrauterine growth [2,3,4]. The misregulation of imprinted gene expression dosage in both murine models and in humans has led to failures of growth regulation, by acting on the intrinsic growth factor signalling pathways, or by regulating intrauterine energy availability via their actions on the development of the placenta. Imprinted genes are also highly expressed in the developing and postnatal brain, where they have been shown to regulate sleep, energy homeostasis and behaviour (reviewed in [1,5]).

From a clinical perspective, imprinted genes are important because their disruption results in a number of paediatric growth/endocrine disorders known as “imprinting disorders” or IDs. IDs such as Silver Russell syndrome (SRS), Beckwith–Weideman syndrome (BWS) and Prader–Willi syndrome (PWS) occur due to various genetic/epigenetic causes such as uniparental disomy (UPD), mutations of genes and ICRs or epimutations that commonly result in an altered IG expression dosage. Such mechanisms can change the expression of multiple genes within a locus; thus, the resultant syndromes are likely to be caused by the aberrant expression of more than one IG. Half a century of careful clinical study of these rare IDs has clarified their paediatric phenotypic spectrum and led to consensus diagnostic criteria (see Table 1 and references therein). In recent years, the accumulation of data from older patients has led to a clearer understanding of lifetime consequences of imprinted gene disruption, and strikingly, postnatal endocrine abnormalities are a common feature of IDs. While not all imprinted genes share their imprinting status between humans and mice, the expression and regulation of the major disease-causing loci are largely conserved [1]. Murine models of imprinted gene disruption are reported to largely recapitulate the human ID phenotypes, particularly with regard to growth and placentation (reviewed in [5]). Less well studied is the action of imprinted genes in the developing and postnatal endocrine system. IGs are highly expressed in the developing and mature hypothalamus, where they play important roles in hormone regulation [5]. However, IGs are also enriched in the pituitary gland (Figure 1), where, to our knowledge, their expression and function has not been systematically examined. Here we attempt to catalogue the expression and potential function of IGs in the pituitary gland, starting with a review of the role and development of that organ.

### 1.1. The Pituitary Gland

The pituitary gland, or hypophysis, is a neuroendocrine organ that regulates many vital physiological functions, including growth, sexual development and reproduction, metabolism, lactation and stress response [43]. Anatomically, the pituitary gland is divided into three lobes: the anterior and the intermediate lobes form the endocrine portion of the hypophysis (adenohypophysis or anterior pituitary), whilst the posterior lobe, together with the infundibulum, constitute the neurohypophysis (or posterior pituitary). The anterior lobe is largely made of five types of hormone-producing cells: somatotrophs, producing the growth hormone (GH), lactotrophs, producing prolactin (PRL), corticotrophs, which produce the adrenocorticotropic hormone (ACTH, or corticotropin), thyrotrophs, releasing the thyroid-stimulating hormone (TSH) and gonadotrophs, responsible for the production of the follicle-stimulating hormone (FSH) and luteinising hormone (LH). A sixth population of hormone-producing cells, the melanotrophs, is found in the intermediate lobe and is responsible for the production of the melanocyte-stimulating hormone (MSH); however, the intermediate lobe appears as a distinct entity in adulthood only in some mammals, such as rodents, whereas in humans, it is present as dispersed cells in the anterior lobe and cyst-lined epithelia between the anterior and posterior lobes (reviewed in [44,45,46,47]). In contrast, the neurohypophysis contains a population of glial cells named pituicytes, and the terminal axonal projections of the magnocellular neurons, responsible for the synthesis of the hormones oxytocin (OT) and vasopressin (or the anti-diuretic hormone, ADH). These hormones are temporarily stored in the posterior pituitary before being released into circulation. Pituicytes are thought to be actively involved in both the storage and release of posterior pituitary hormones [48,49,50].

#### 1.1.1. Function

The pituitary gland has a central role in regulating the function of target organs. However, the activity of the pituitary gland is itself finely regulated by the hypothalamus through the production of neuropeptides acting on the pituitary hormone-producing cells. Thus, it is probably more accurate to define the hypothalamic-pituitary (HP) axis as the central endocrine regulator of the body. The most abundant cell type of the adenohypophysis are the somatotrophs [44], which produce GH in response to the hypothalamic GH-releasing hormone (GHRH) and ghrelin, whilst being negatively regulated by somatostatin. The growth hormone/insulin-like growth factor 1 (GH-IGF1) pathway is the major endocrine growth pathway in mammals. GH stimulates growth and metabolism by directly targeting bone, muscular and adipose tissue, as well as by inducing the production of hepatic insulin-like growth factor 1 (IGF1). Circulating IGF1 autoregulates GH production by a negative feedback mechanism to the IGF1 receptors on the somatostatin/GHRH neurons and the somatotrophs (reviewed in [51]). The tuberoinfundibular dopaminergic (TIDA) neurons in the hypothalamic dorsomedial arcuate nucleus produce dopamine, a strong inhibitor of lactotroph function. In the absence of dopamine, these cells produce PRL, mostly known for stimulating mammary gland development and lactation (further discussed in this paper and reviewed in [52]). TSH production by thyrotrophs, the smallest hormone-producing population of the anterior pituitary, is positively regulated by the hypothalamic TSH-releasing hormone (TRH). As the name suggests, TSH stimulates the follicular cells of the thyroid gland to produce thyroxine (T_4_) and triiodothyronine (T_3_), key regulators of the body’s metabolic rate (reviewed in [53]). FSH and LH are crucial for the sexual development, maturation and reproduction of both females and males. They are produced by the gonadotrophs, which are positively stimulated by the hypothalamic gonadotropin-releasing hormone (GnRH) (reviewed in [54]). Different cleavages of the precursor polypeptide pro-opiomelanocortin (POMC) give rise to ACTH and α-MSH in corticotrophs and melanotrophs, respectively. ACTH stimulates the adrenal cortex to produce glucocorticoids. The hypothalamus positively regulates this process through the production of the corticotropin-releasing hormone (CRH) (reviewed in [55]). Additionally, the target organs at the end of these endocrine axes can signal to the pituitary and the hypothalamus to control their hormonal production, giving rise to finely regulated feedback mechanisms.

#### 1.1.2. Development

The duality of the pituitary gland is reflected in its dual embryonic origin, with the anterior and posterior pituitary arising from the oral ectoderm and neural ectoderm, respectively. Much of what we know about the development of this organ has come from embryological-genetic studies performed in rodents. In mice, pituitary organogenesis begins around embryonic day (e)8.5, with a thickening and invagination of the oral ectoderm that forms the pituitary primordium, also known as the Rathke’s pouch (RP). By e9.5, this rudimentary pouch is in close contact with the ventral diencephalon, which regulates pituitary development through the production and secretion of key signalling factors, including members of the bone morphogenic protein (BMP), fibroblast growth factor (FGF), Sonic Hedgehog (SHH), NOTCH and WNT signalling pathways [56,57,58,59,60,61,62]. 

By e11.5, the RP appears as a defined structure that largely consists of a central lumen surrounded by highly proliferative epithelial progenitors expressing the stem cell marker, sex-determining region Y-related HMG box 2 (SOX2) [63]. At the same time, the posterior lobe begins to form from an evagination of the ventral diencephalon. Between e11.5 and e13.5, the SOX2^+^ progenitors gradually exit the mitotic cycle and start to express specific lineage commitment markers that allow for the formation of the distinct pituitary cell types [64]. Once the differentiation process has begun, some progenitors migrate ventrally and rostrally, giving rise to the anterior lobe (AL), whilst the cells located in the dorsal part of the RP form the intermediate lobe (IL) [65]. The corticotrophs are the first cells to appear at e12.5 and reach terminal differentiation. Initially, these cells require the activation of the T-box transcription factor TPIT (TBX19), which is a pioneer factor also required for melanotroph differentiation [66,67]. For this reason, corticotrophs and melanotrophs (both expressing the *Pomc* gene) are classified as the TPIT-lineage. Subsequently, cells that will become corticotrophs transiently express the bHLH transcription factor NeuroD1, while the expression of the paired-box protein PAX7 induces melanotroph specification [68,69]. Somatotrophs, lactotrophs and thyrotrophs all originate from progenitors expressing the POU-homeodomain protein known as pituitary-specific positive transcription factor 1 (PIT1), hence constituting the PIT1-lineage. PIT1 starts to appear at e13.5 and its expression is activated by the paired-homeodomain transcription factor Prophet of PIT1 (PROP1), which is the earliest pituitary-specific transcription factor to be expressed during pituitary development, around e10 [70,71,72,73,74]. Terminal differentiation of the hormone-producing cells from the PIT1-lineage is achieved through the expression of specific factors, the bHLH factor NeuroD4 for the somatotrophs, the transcription factors forkhead box protein (FOX), L2 and GATA2 for the thyrotrophs, and oestrogen receptors for the lactotrophs [59,71,75,76,77,78]. The last lineage to appear at e16 is the gonadotroph lineage, which comprises LH- and FSH-producing cells [65]. Progenitor cells that become committed to the gonadotroph lineage start to express the GnRH receptor from e12.5, followed by expression of the steroidogenic factor 1 (SF1/NR5A1) at e14.5 [79,80]. 

In mice, the anterior and intermediate lobes that constitute the anterior pituitary are separated by a luminal space known as the cleft. Notably, the periluminal region (also known as marginal zone, MZ) is described as the stem cell niche of the pituitary gland [63,81,82]. During the early stages of pituitary organogenesis, Rathke’s pouch, the anterior pituitary primordium, is almost entirely made of SOX2^+^ progenitors. At later stages of development and in adulthood, SOX2^+^ PSCs are located in the marginal zone along the cleft, as well as in clusters throughout the AL parenchyma [83]. From e14.5, a proportion of PSCs start to express another transcription factor, SOX9, and by the time pituitary development is completed, SOX2^+^SOX9^+^ double-positive cells represent the majority of PSCs [63,81]. This population has the ability to self-renew and to give rise to terminally differentiated pituitary cells, in vitro and in vivo [63,81,82,84], and in response to injury [85]. Additionally, they secrete paracrine signalling factors, including WNT ligands, which promote the generation of new endocrine cells from more committed neighbouring progenitors [86]. Thus, adult progenitor/stem cells appear to contribute to the plasticity of the pituitary gland both directly and indirectly, enabling adaptation and generation of new endocrine cells in response to physiological events or injuries. Nonetheless, most of the adult PSCs will remain quiescent throughout life [81,82].

#### 1.1.3. Endocrine Aspects of Human Imprinting Disorders

Human imprinting disorders (IDs) are a set of syndromes that have multiple endocrine involvement [87]. Diagnosis of IDs in early life is often complicated by the high degree of phenotypic overlap—particularly associated with intrauterine growth restriction (IUGR) (transient neonatal diabetes mellitus (TNDM), SRS, Temple syndrome (TS)), perinatal hypotonia and failure to thrive (SRS, TS, PWS, see Table 1 and references therein). Further, multiple imprinted gene-disease clusters can be affected in patients carrying mutations in genes encoding trans-regulators of the imprinting mechanism (multi-locus imprinting disturbances, or MLIDs [88]). Taken together, common pathways of endocrine physiology influenced by imprinted genes emerge—the regulation of postnatal growth, establishment of independent feeding and timing of puberty. However, since these processes are highly interdependent, understanding the molecular physiology underlying IDs is challenging, but likely to yield important insights into the mechanisms and evolutionary trade-offs that drive growth and reproductive maturity.

SRS, BWS, TS and PWS are all associated with postnatal growth abnormalities (Table 1). SRS and TS patients experience growth restriction and feeding difficulties. While GH deficiency is not a feature of either syndrome, there is evidence supporting changes in GH action in SRS patients. GH treatment is part of the clinical response to SRS, and most children are treated in order to improve muscle mass and appetite, as well as to increase growth velocity. However, some SRS patients, particularly those with epimutations at 11p15.5, exhibit very high serum IGF1 and IGF-binding protein 3 (IGFBP3) levels in response to GH. Moreover, compared to other children born small for their gestational age, SRS patients attain a reduced final height with comparable GH treatment, suggesting IGF1 resistance [11]. BWS is associated with pre- and postnatal macrosomia, though the relative contribution of each growth interval to the eventual overgrowth is dependent upon the underlying epi/genetic cause. Patients with imprinting centre 1 gain of methylation (IC1-GoM) tend to be large at birth with less frequent postnatal overgrowth compared to patients with imprinting centre 2 loss of methylation (IC2-LoM) or mutations in cyclin-dependent kinase inhibitor 1C (*CDKN1C*), who are more likely to have a birthweight in the normal range, but also more likely to experience increased postnatal growth velocity [89]. This suggests an insulin-like growth factor 2 *(IGF2*)-mediated prenatal overgrowth mechanism that is somewhat separable from the postnatal overgrowth driven by *CDKN1C* loss of function. Consistently, patients with SRS due to *IGF2* deletion are born small but have normal growth velocity after birth [12]. PWS patients have a short stature, reduced bone density and low GH and IGF1 levels, consistent with primary GH deficiency [31]. PWS is caused by loss of expression of the paternally expressed genes at 15q11-q13. Loss of function mutations in one such gene, melanoma antigen, family L, member 2 (*MAGEL2*), cause Shaaf–Yang syndrome, which has considerable phenotypic overlap with PWS in infancy [38]. Notably, the loss of *MAGEL2* function appears to cause at least some of the endocrine abnormalities of PWS, since patients with mutations in this gene display panhypopituitarism associated with a hypoplastic anterior pituitary gland [39]. Finally, acromegaly is associated with impaired expression of the guanine nucleotide-binding protein (*GNAS*). McCune–Albright syndrome is caused by maternal inheritance of *GNAS* gain-of-function mutations and associated with hyperfunction endocrinopathies, including GH excess [41].

SRS, TS and PWS are all associated with impairments in gonadal development and/or alterations to the timing of puberty (summarised in Table 1). Some SRS patients experience premature adrenarche, with early and rapid puberty (reviewed in [11]). The maternal inheritance of *CDKN1C* gain-of-function mutations causes IMAGe syndrome, or intrauterine growth restriction, metaphyseal dysplasia, adrenal hypoplasia congenita and genital anomalies, which shares a phenotypic overlap with SRS [26]. This condition can manifest with hypogonadotropic hypogonadism, but only in males. The majority of the endocrine abnormalities in these patients are likely to be due to developmental adrenal hypoplasia, rather than pituitary dysfunction, since ACTH feedback regulation appears normal [90]. A second gene in the IC2 region, potassium voltage-gated channel subfamily Q member 1 (*KCNQ1*), has also been associated with gonadal development, since maternally inherited missense mutations in this locus can cause growth hormone deficiency (GHD) and gingival fibromatosis (OMIM 611010), a condition that presents with gonadotrophin deficiency [27]. TS is caused by the misregulation of imprinted genes on chromosome 14q32 and is characterised by IUGR with postnatal hypotonia proceeding to childhood truncal obesity, short stature and premature sexual development [28]. Recently, loss-of-function mutations in the paternally expressed delta-like homologue-1 gene (*DLK1*), located at 14q32, have been linked to central precocious puberty (CPP) [30]. Hypogonadism is an important clinical feature of PWS (reviewed in [32]). Loss of *MAGEL2* function probably contributes to this pathophysiology, since loss-of-function mutations can cause hypopituitarism and gonadotrophin deficiency [39]. A second gene in the PWS cluster, makorin ring-finger protein 3 (*MKRN3*), may also contribute to this phenotype, since the paternal inheritance of inactivating mutations causes CPP [40]. However, how PWS cluster genes might interact to contribute to the pubertal phenotype remains unclear. Finally, imprinted genes have been implicated more widely as factors that account for normal genetic variation in pubertal timing. A large genome-wide association study for age at menarche identified variants at four imprinted regions (*DLK1*, *MKRN3*, *MAGEL2* and *KCNK9,* or potassium two-pore-domain channel subfamily K member 9) that significantly impact pubertal timing [91].

Because of the hypothalamic expression pattern of some of the imprinted genes involved in postnatal growth and gonadotrophin regulation, it has been assumed that imprinted genes influence these pathways at the level of the central nervous system (CNS) by controlling neuronal development and connectivity [92]. However, many imprinted genes have expression in the developing and postnatal pituitary gland (see below). We argue that there may be additional roles for imprinted genes in the development and function of the pituitary gland that are relevant to normal endocrine physiology and human imprinting disorders.

#### 1.1.4. Co-Ordinate Regulation of Imprinted Genes

The commonality of phenotypes between different IDs and murine imprinted gene knock-out models suggested that imprinted gene products may be linked into molecular and regulatory pathways. Numerous studies have now demonstrated that imprinted genes from disparate chromosomal clusters are co-regulated [93,94]. Experiments utilising a meta-analysis of multiple transcriptomic datasets [94] as well as direct manipulations of cell lines [95] showed that a subset of ~15 imprinted genes are co-regulated (collectively known as the imprinted gene network, or IGN), with their expression peaking in cycling cells at the transition between proliferation and quiescence. In this context, dynamic changes to imprinted gene expression occur in the absence of changes to the level of CpG methylation at imprinting control regions [95]. Rather, the co-ordinate control is thought to be mediated by trans-acting factors, some of which are themselves imprinted, including the transcription factor pleiomorphic adenoma gene-like 1 (PLAGL1) [96], and the non-coding RNAs *H19* [97] and *IPW* (imprinted in PWS [98]).

IGN genes are downregulated in vivo in multiple somatic tissues, concomitant with growth deceleration in the early postnatal period of the mouse [99]. In addition, the IGN is regulated by the insulin/insulin-like growth factor pathway during adipogenesis [100] and associated with an epigenetically regulated biphasic obesity in mice and humans [101]. Importantly, several members of the IGN are expressed in somatic stem cell niches and contribute to postnatal tissue homeostasis, including *Igf2* [102], *Dlk1* [103], *H19* [104] and paternally expressed gene 3 (*Peg3*) [105,106]. These observations have led to the idea that the IGN might generally define somatic stem cell populations and be necessary for adult tissue regeneration [107]. The murine pituitary gland is an excellent system in which to study adult stem cell dynamics, since the SOX2-stem/progenitor cell population is well defined and histologically distinct [81,82]. To our knowledge, the role of the IGN in pituitary stem cells has not yet been explored.

#### 1.1.5. Broader Roles for Imprinted Genes in Maternal-Offspring Communication

Pregnancy is a time of profound endocrine disturbance for the mother. Hormones produced by the placenta enter the maternal circulation and act on many aspects of maternal physiology in order to support the foetus and ready the mother for birth and lactation. Many of these hormones influence the maternal hypothalamo-pituitary feedback mechanisms. Placental lactogens, oestrogen and progesterone are present at high concentrations in maternal blood from mid-gestation. These hormones act to suppress normal ovarian cycling through the hypothalamic gonadotrophin system [54] and to prepare the mother for lactation through the prolactin pathway (reviewed in [108]).

Hormones of pregnancy have profound and dynamic effects on maternal pituitary PRL secretion; in early pregnancy in rodents, pituitary PRL is released in a pattern of twice-daily surges. The onset of high placental lactogen production at mid-gestation suppresses maternal PRL until approximately 24 h before parturition, when maternal PRL is restored to high levels, remaining so for the duration of the lactation period [108]. Pituitary PRL production is controlled by neuroendocrine dopaminergic circuits, most notably the TIDA neurons, which integrate multiple neuroendocrine inputs (reviewed in [109]). TIDA neurons arise from the hypothalamic arcuate nucleus and project to the median eminence, which connects to portal blood vessels that contact the lactotroph cells. Additional neuronal pathways interconnect with the intermediate and posterior lobes of the pituitary gland (tuberohypophyseal and periventricular hypophyseal neurons). Dopamine is secreted from these neurons and acts on D2 dopamine receptors on lactotrophs to suppress PRL secretion. PRL secretion is regulated by a pituitary-hypothalamic feedback loop; PRL enters the brain, likely through a transport system involving PRL receptors in the choroid plexus epithelium. In the brain, PRL stimulates the production of tyrosine hydroxylase (TH), the rate-limiting step in the production of dopamine [109]. In mid- to late pregnancy, placental lactogens, oestrogen and progesterone act on TIDA neurons, and thus, suppress maternal pituitary PRL secretion [108].

In rats and humans, a rise in pregnancy-associated oestrogen is thought to promote an increase in the lactotroph number in preparation for lactation [110,111]. In mice, the lactotroph number does not increase in pregnancy, but rather, there is an increase in cell size [112]. Pregnancy is also associated with the increased secretion of non-placental GH in rats and mice [113,114]. This increase is not due to the elevated transcription of *Gh*, but rather due to increased pituitary secretory activity, GH stability and possibly an extra-pituitary source [113].

The function of imprinted genes is closely connected to embryonic growth and maternal reproductive success (reviewed in [115]). As well as directly influencing embryonic and placental growth pathways, it is now well recognised that imprinted genes can influence gestational hormone production. Notably, imprinted gene dosage manipulation in the mouse can alter the size and activity of the placental endocrine compartment to influence the production of placental lactogens (reviewed in [116]), and foetal production of DLK1 promotes maximal maternal GH levels circulating in pregnancy [114]. Despite this, little attention has been paid to how imprinted genes might act in the mother to influence her sensitivity to endocrine manipulation by the conceptus. There is some evidence that such a mechanism may be occurring during the lactation period, since dams lacking a functional maternal allele of growth factor receptor-bound protein 10 (*Grb10*), exhibit PRL resistance [117]. In addition, several imprinted gene knock-outs in mice influence latency to adopt maternal behaviour, a trait partially controlled by PRL signalling [109,116].

Additional evidence suggests that maternal imprinted gene expression in the pituitary gland might be influenced by pregnancy. Female virgin rats treated with oestradiol (E2) for three weeks experienced a threefold increase in pituitary weight and a twofold increase in *PRL* gene expression. Pituitary transcriptomic data from this study indicated that the imprinted gene network may be upregulated by elevated oestrogen in pregnancy. Compared to virgin rats, the expression levels of *PLAGL1*, *CDKN1C*, insulin-like growth factor 2 receptor (*IGF2R*), *DLK1* and decorin (*DCN*), were altered following E2 treatment [118]. Taken together, these data suggest that in the maternal pituitary gland, the imprinted gene network might constitute a target of the hormones of pregnancy.

### 1.2. Aims of the Study

In this work we utilise previously published transcriptomic datasets to provide a comprehensive description of the expression of imprinted genes in the murine pituitary gland. In addition, we present novel transcriptomic data from the embryonic and the pregnant pituitary gland. We relate our findings to previous work describing murine manipulations of imprinted genes and the endocrine phenotypes observed in human imprinting disorders. Finally, by comparing the transcriptomes of the mid-gestation maternal pituitary gland to that of virgin females, we seek to understand to what extent the expression of imprinted genes in the maternal pituitary gland is influenced by pregnancy, a major disruptor of pituitary hormone homeostasis.

## 2. Materials and Methods

### 2.1. Mice

For mice used in the embryo vs. the adult comparison, the animals were bred on a mixed 129Sv/c57BL6J genetic background. All other animals were maintained on an inbred c57BL6J background. Mice were housed in a temperature- and humidity-controlled room with a 12 h/12 h light-dark cycle. All mice were fed ad libitum, given fresh tap water daily and re-housed in clean cages weekly. Mice were weaned at 21 days postnatum, or a few days later if they were particularly small. Thereafter, they were housed in single-sex groups (5 per cage maximum) or occasionally singly housed. Embryos were collected from timed matings, where embryonic day (e) denotes days following the detection of a copulation plug.

### 2.2. Embryo vs. Adult Comparison

The curated imprinted gene list (Appendix A) was based on the Geneimprint (https://www.geneimprint.com, accessed on 4 July 2020) and MouseBook (https://www.mousebook.org/imprinting-gene-list, accessed on 4 July 2020) catalogues, as well as in a recently published list [119].

Rathke’s pouch was dissected from 7 *Sox2^Egfp/+^* [120] e13.5 embryos (2 male, 5 female). The whole pituitary gland was dissected from 2 adult mice at postnatal week 8 (1 male, 1 female) and the posterior lobe was removed. Samples were collected in TRIzol^TM^ (Thermo Fisher: Waltham, MA, USA) and kept at −80 °C until further processed. RNA was prepared following the manufacturer’s protocol with the following modifications: The aqueous solution was precipitated overnight using isopropanol and 10 μg of glycogen (Ambion) to increase the recovery of RNA. Samples were centrifugated at 12,000 *g* for 45 min, washed twice with 70% EtOH and RNA was dissolved in RNAse-free water (Qiagen). Samples were quantified using QuBit fluorometers (Thermo Fisher: Waltham MA, USA) and RNA purity was measured using a Nanodrop. A Bioanalyzer RNA 6000 Pico assay (Agilent Technologies, Santa Clara, CA, USA) was used to check the RNA quality. RNA samples with an RNA integrity number (RIN) above 9 (9–9.5) were selected and sent to the Oxford Genomic Centre (Oxford, UK) for the RNAseq experiment. Samples were normalised to a 50 ng input. The purification of mRNA, generation of double-stranded cDNA and library construction were performed using the NEBNext Poly(A) mRNA Magnetic Isolation Module (E7490) and the NEBNext Ultra II Directional RNA Library Prep Kit for Illumina (E7760L) with the centre’s adapters and barcode tags (dual indexing [121]), before sequencing on an Illumina HiSeq 4000 as 75 bp paired end.

Raw sequencing reads were trimmed using the Trimmomatic v0.39 [122] with the default settings and then mapped to the mm10 reference genome using STAR (Galaxy Version 2.7.5b) [123]. Aligned reads were counted over genes (GRCm38 annotations downloaded from Ensembl version 96) using the “--quantMode GeneCounts” option of STAR. Gene counts were normalised using the DESeq2 R/Bioconductor package (v1.30.0) [124]. After excluding genes with mean read count < 5 across all samples, the DESeq2 was used to assess differential gene expression between the embryo and adult pituitary samples. Next, *p*-values were adjusted by the Benjamini–Hochberg procedure for controlling the false discovery rate in multiple comparisons, and genes with a *p*-value < 0.05 and fold change > 4 were considered to be significantly differentially expressed. The heatmaps show a regularised log transformation implemented in the DESeq2 R/Bioconductor package (v1.30.0) [124], and applied to normalised counts and the values, which were then z-transformed across samples.

### 2.3. Single-Cell RNA Sequencing (scRNAseq) Data Processing

Single-cell sequencing data from the mouse pituitary gland was downloaded from GEO using accession numbers GSE120410 (postnatal day 4, P4) and GSE142074 (postnatal day 49, P49). Using Seurat (v3.1.5) [125,126] in R, cells from both datasets were taken forward if they expressed between 1000–5500 genes and <10% mitochondrial transcripts, removing doublets and low-quality cells. Transcripts expressed in >3 cells were included for downstream analysis. To evaluate the two datasets together, filtered cells were integrated using the SCTransform workflow [127]. Clustering and visualisation for the integrated objects were carried out using 0.5 resolution and 1:12 principal components. Clusters were named according to known cell-type markers as previously reported [44]. Four somatotroph clusters and two lactotroph clusters were found, which were respectively grouped together and named accordingly before the downstream analysis. The identification of differentially expressed (DE) imprinted genes (IGs) was carried out using the “FindAllMarkers” function in Seurat on the “RNA” assay of the combined object and subset of P4 and P49, respectively, with input features of all known IGs (Appendix A). Significant DE IGs were reported and taken forward from transcripts with *p*_val_adj ≤ 0.05 and avg_logFC ≥ 1. Plots of specific transcript expression were created using the “FeaturePlot” function in Seurat with a min.cutoff = 0. To find cells expressing specific IGs, the “WhichCells” function in Seurat was used, and cells were counted as positive if expression was >0. Venn diagrams to visualise the overlap of cells positive for specific IGs were made using Eulerr [128].

### 2.4. Pregnant vs. Virgin Comparison

8-week-old c57BL6/J females from 4 independent litters were randomly allocated to the mating (*n* = 8) or virgin (*n* = 4) groups. Females were mated with c57BL6/J studs, the day of the vaginal plug was recorded and the gravid females were sacrificed at e15.5 by terminal anaesthesia. Pregnant females carried between 5–8 embryos/litter (median = 7). The whole pituitary was removed from the skull and immediately snap frozen, then stored at −80 °C.

RNA for microarray analysis was extracted using a Qiagen RNA extraction kit, including the DNase step. The total RNA quality was verified using an Agilent 2200 TapeStation and concentrations were measured by the Qubit fluorometer. Labelled targets were generated from the total RNA and hybridised to the MouseWG-6 v2.0 Expression BeadChip (Genomics Centre, KCL, UK). Using the Lumi package in Bioconductor [129], the data were quantile normalised and processed with variance stabilizing transformation. Differential expression between the pregnant and non-pregnant animals was undertaken using Limma in Bioconductor [130] and empirical Bayes statistics were calculated. Genes were considered significantly differentially expressed if they exhibited at least a 1.25-fold difference in expression with a false discovery rate (FDR) less than 0.05. Enrichment in gene ontology (GO) biological process terms for differentially expressed genes was assessed using the cluster Profiler package [131]. Significant GO terms were defined at FDR < 0.05.

Validation of DE targets was performed on an independent cohort of c57BL6/J pregnant and virgin mice at 8–12 weeks of age. The total RNA from the whole pituitaries (9 pregnant, 7 virgin) was prepared using TRIzol^TM^ (Thermo Fisher), as described above. The RNA samples were treated with DNase I (M0303, New England Biolabs (NEB)), following the manufacturer’s instructions. Complementary DNA (cDNA) was obtained by reverse transcription (RT) using 1 μg of purified RNA as a template and Moloney murine leukaemia virus (M-MuLV) reverse transcriptase (M0253, NEB), using the standard first strand synthesis protocol with random hexamers (S1230, NEB), 0.625 mM dNTPs and RNase inhibitors (M0314, NEB). A negative control to test for genomic contamination was run alongside (“-RT sample”). Samples were diluted 20× before being analysed using SYBR Green real-time quantitative PCR (RT-qPCR; QuantiTect SYBR Green PCR kit, Qiagen). Quantification was performed as previously described [114]. The mean of *Actb* and *α-tubulin* expression was used to normalise the expression of the target genes. Details of the primers used in this study are included in Appendix A.

### 2.5. Histology

Fresh embryos and postnatal pituitaries were fixed in 4% *w*/*v* paraformaldehyde (PFA, P6148, Millipore-SIGMA: Burlington, MA, USA) in phosphate-buffered saline (PBS, BR0014G, Thermo Scientific Oxoid) overnight and dehydrated through an increasing ethanol series. Samples were stored at 4 °C in 70% ethanol or dehydrated to 100% ethanol before being processed for paraffin embedding. The day of the embedding, the samples were incubated at room temperature with Histoclear (National Diagnostics; 2 × 20 min for e9.5–e11.5 and postnatal pituitaries, 2 × 35 min for e13.5) or Xylene (VWR; 2 × 45 min for e15.5, 2 × 1 h for e18.5). This was followed by 3 × 1-h incubations at 65 °C with Histosec^®^ (1.15161.2504, VWR). Histological sections of 5 μm were used for in situ hybridisation (ISH) and immunohistochemistry (IHC).

### 2.6. In Situ Hybridisation

Expression analyses on histological sections were carried out using ISH when working antibodies for the selected targets were not available. ISH was performed as previously described [132]. Sections were hybridised overnight at 65 °C with sense and antisense digoxigenin (DIG)-riboprobes against *Igf2* [133]*, Cdkn1c* and neuronatin (*Nnat*; details shown in Appendix A). Sections were washed and incubated overnight at 4 °C with an anti-digoxigenin-AP antibody (#45-11093274910 Millipore-SIGMA, 1:1000). Staining was achieved by adding a solution of 4-Nitro blue tetrazolium chloride (NBT, 11383213001, Roche) and 5-Bromo-4-chloro-3-indolyl phosphate disodium salt (BCIP, 11383221001, Roche). The sections were mildly counterstained with Nuclear Fast Red (H-3403-500, Vector Laboratories) and mounted using dibutyl phthalate in xylene (DPX, 6522, Millipore-SIGMA). Sense controls for each probe were tested at e13.5 and showed no staining under identical conditions (Appendix A).

### 2.7. Immunohistochemistry

IHC on histological sections was performed as previously described [82,132]. Detection of the proteins was achieved by incubating the histological sections overnight at 4 °C with the following primary antibodies: rabbit α-growth-factor receptor bound protein 10 (GRB10) (PA5-79322, Invitrogen, 1:200), rabbit α-SOX2 (ab92494, Abcam, 1:300), mouse α-DOPA decarboxylase (DDC) (sc-293287, Santa Cruz, 1:200). The following day, the slides were incubated for 1 h at room temperature with secondary biotinylated goat α-rabbit or goat α-mouse (BA-1000 and BA-9200, Vector Laboratories, 1:300), followed by a 1-h incubation at room temperature with the Vectastain^®^ Elite ABC-HRP kit (PK-6100, Vector Laboratories). Staining was achieved through colorimetric reaction using the DAB peroxidase substrate kit (SK-4100, Vector Laboratories). Slides were mildly counterstained with Mayer’s haematoxylin (MHS16, Millipore-SIGMA) and mounted using DPX. The secondary antibody only controls for each antibody, and showed no staining under identical conditions (Appendix A).

## 3. Results

### 3.1. Imprinted Genes Are Highly Expressed in the Anterior Pituitary Gland and Are Developmentally Regulated

Imprinted genes are known for their actions on embryonic growth, and many are highly expressed in developing tissues and their expression declines postnatally [99] (reviewed in [134]). To determine if this temporal behaviour is also true for imprinted genes in the pituitary gland, we generated transcriptomic datasets by bulk RNA sequencing from the developing and adult pituitary (Figure 1A). At e13.5, Rathke’s pouch had fully invaginated from the oral epithelium. Proliferating SOX2^+^ progenitor cells formed the bulk of the cell population, but some cells were beginning to commit towards hormonal lineages. Conversely, in the adult gland, all hormonal cell types were represented and SOX2^+^ cells formed a minority of the cell population [81,82].

Imprinted genes were some of the most highly expressed of all genes in this organ, both during development and in the mature gland (Figure 1B). When comparing embryonic and adult transcriptomes, we found that imprinted genes as a group were enriched amongst the differentially expressed genes (Figure 1C). This included genes that were more highly expressed in the embryo (“embryo high”, 18/111, 16%), and those which had more abundant transcripts in the adult (“adult high”, 24/111, 21%) (Figure 1D and Appendix A). Since imprinted genes are often found in genomic clusters that can be regulated by shared enhancers [135], we predicted that temporal gene expression patterns would be shared within imprinting clusters. However, with the exception of the *Igf2*-*H19* cluster, we did not find strong evidence of co-ordinated temporal gene regulation (Appendix A).

Among the “embryo high genes”, half overlapped with those previously identified as postnatally downregulated by Lui and colleagues (*Grb10*, *H19*, *Igf2*, *Cdkn1c* and solute carrier family 38, member 4 (*Slc38a4*) [99]). The expression of these genes could be associated with the proliferating progenitor population that diminishes with maturity. We explored this idea by comparing the developmental expression pattern of four “embryo high” genes (*Igf2*, *Cdkn1c*, neuronatin (*Nnat*), and *Grb10*) with that of the progenitor marker SOX2 (Figure 2). *Igf2*, *Cdkn1c* and *Nnat* mRNA, as well as the GRB10 protein, were expressed at high levels in RP at e11.5 (Figure 2A,H,O,V), co-incident with the expression of SOX2 (Figure 2Ad). As development proceeded, the detection of *Igf2*, *Cdkn1c* and GRB10 was reduced, whereas *Nnat* expression was retained in the majority of anterior pituitary (AP) cells (Figure 2S–U). In the mature pituitary, *Igf2* and *Cdkn1c* expression was undetectable (Figure 2G,N), whereas GRB10 could be observed at low levels in postnatal cells in both the parenchyma of the AP and in the posterior lobe (Figure 2Ac).

### 3.2. “Embryo High” Imprinted Gene Expression Is Not Retained in the Postnatal Stem Cell Population

In order to fully understand the expression patterns of imprinted genes in the postnatal pituitary gland, we combined and reanalysed two previously published single-cell RNA sequencing (scRNAseq) datasets generated by the Camper lab [44,136] (Figure 3A). We confirmed that our analysis produced biologically meaningful clustering by overlaying the expression of markers of hormone-producing cells and stem/proliferating populations, as well as minority cell types such as endothelial cells and connective tissue (Appendix A). The majority of imprinted genes (100) had detectable expression in the postnatal scRNAseq data, and the frequency and level of their expression is summarised in Figure 3B.

Using these data, we first asked: is expression of “embryo high” imprinted genes that are postnatally repressed in the whole pituitary retained in the stem cell population? Of the 18 “embryo high” imprinted genes, none were found to be enriched in stem cells. However, imprinted genes were found amongst the genes with significantly enriched expression in the postnatal pituitary stem cell cluster at both postnatal day 4 and day 49 (Appendix A and Table 2), including IGN members *Plagl1*, *Peg3* and *Igf2r*. The expression of *Plagl1* and zinc finger DBF-type containing 2 (*Zdbf2*), was additionally enriched in proliferating cells, indicating that the expression of these genes may be associated with active stem cells that are maintaining organ homeostasis.

**Table 2 genes-12-00509-t002:** Enriched imprinted genes in single-cell RNA sequencing (scRNAseq) postnatal AP cell types. Identification of differentially expressed (DE) imprinted genes (IGs) was carried out on the combined-age data (postnatal day 4 (P4) and postnatal day 49 (P49) together), and a subset of P4 and P49, respectively. Significant DE IGs were reported if *p*_val_adj ≤ 0.05 and avg_logFC ≥ 1.

Cell Type	Combined P4 & P49	P4	P49
Stem Cells	*Igf2r, Pdk4, Kcnq1ot1, Zim1, Peg3, Sgce, Plagl1, Zdbf2, Usp29, Ube3a*	*Gab1, Igf2r, Peg3, Zim1, Kcnq1ot1, Pdk4, Plagl1, Zdbf2*	*Pdk4, Pon2, Sgce, Plagl1, Kcnq1ot1*
Proliferating Cells	*Plagl1, Commd1, Zdbf2, Tssc4, Mdh2*	*Tssc4, Mdh2, Commd1, Zdbf2*	*Tssc4, Commd1, Mdh2, Zdbf2, Ube3a*
Somatotrophs	*Dlk1*		*Dlk1*
Lactotrophs	*Meg3, Blcap, H13, Cdkn1c, Nap1l5*		*Nnat, Asb4, Meg3, Blcap, H13, Nap1l5*
Thyrotrophs	*Peg10, Gnas, Mcts2, Snrpn*	*Dlk1, Gnas, Peg10, Nap1l5, Snrpn, Mcts2*	
Melanotrophs	*Usp29*	*Usp29*	
Corticotrophs	*Usp29*	*Nap1l5, Usp29, Magi2, Impact*	*Usp29*
Sf1 progenitors	*Bcl2l1*	*Bcl2l1, H13, Blcap*	*Cdkn1c*
Gonadotrophs	*Qpct, Rasgrf1, Ndn, Nnat*	*Qpct, Th, Nnat*	*Rasgrf1, Qpct, Gnas, Snrpn, Ndn, Kcnq1ot1, Mdh2*

### 3.3. Clustered Imprinted Genes Show Overlapping Cell-Specific Gene Expression

The expression of some imprinted genes was enriched in distinct hormone-producing cell populations (Figure 3B–E and Table 2). Bladder cancer-associated protein (*Blcap*) and *Nnat*, a gene-retrogene pair [137], appeared co-expressed and enriched in the lactotrophs, with some additional expression in the gonadotrophs and corticotrophs. *Asb4*, or ankyrin repeat and SOCS box-containing 4, and maternally expressed gene 3 (*Meg3*) expression was also enriched in lactotrophs (Figure 3C). PWS cluster genes small nuclear riboprotein polypeptide *n* (*Snrpn*) and necdin (*Ndn*), had overlapping expression distribution and their expression was significantly enriched in gonadotrophs, though both were expressed in additional hormonal cell types. Other gonadotroph markers included ras protein-specific guanine nucleotide releasing factor 1 (*Rasgrf1)* and glutamy-peptide cyclotransferase (*Qpct*; Figure 3D). Finally, the *Peg3* cluster genes appeared to have a high degree of similarity of gene expression in Uniform Manifold Approximation and Projection (UMAP) plots—with significantly enriched expression of *Peg3*, zinc finger imprinted 1 (*Zim1*) and ubiquitin-specific peptidase 29 (*Usp29*) in stem cells and high expression in ACTH-producing corticotrophs and melanotrophs, particularly at P4 (Table 2, Figure 3B,E).

### 3.4. Imprinted Gene Expression in the Pituitary Gland of the Pregnant Dam

Since imprinting is associated with maternal-offspring interactions in the context of endocrine-metabolic adaptations [115], we asked if imprinted genes as a class were modulated in the pregnant pituitary gland. We compared transcriptomes of whole pituitary glands between wild-type virgin female mice and their pregnant littermates in late gestation (at e15.5, Figure 4A). The state of pregnancy robustly modified the pituitary transcriptome, explaining the second largest component of the variation between samples (Figure 4B). We identified 190 genes which were differentially expressed between groups, 103 were upregulated and 87 were downregulated in pregnant dams compared to virgin littermates (Figure 4C). Gene-set enrichment analysis highlighted multiple significantly enriched terms (Appendix A), including expected categories linked to vascular remodelling (e.g., GO:0048659 smooth muscle cell proliferation) and neuroendocrine signalling (GO:0099601 regulation of neurotransmitter receptor activity). As expected, amongst the most downregulated genes in the pregnant pituitary were components of the gonadotrophin pathway, which is suppressed by the hormones of pregnancy (Appendix A, e.g., luteinising hormone β subunit (*Lhb*), early growth response 1 (*Egr1*), and AP1 transcription factors, [138]). Amongst the most upregulated genes were *Igfbp3* and pappalysin 2 (*Pappa2*), which regulate local IGF1 bioavailability ([139], Appendix A).

Imprinted genes as a class were not modified by the condition of pregnancy. Of the 190 genes that were differentially expressed between the pregnant dams and virgin littermates, only three were subject to genomic imprinting: *Qpct* [140], *Ddc* [141] and *Grb10* [142]. Moreover, all three genes are imprinted in a tissue-specific manner, but their imprinting status in the pituitary gland has not yet been established. However, we observed in our adult RNAseq data that *Grb10* and *Ddc* were both expressed from alternative promoters that give rise to imprinted transcripts, *Grb10* from the maternally inherited alleles and *Ddc* from the paternally inherited alleles (Appendix A [141,143]). We confirmed the relative upregulation of *Ddc* and *Grb10* in an independent cohort of age-matched virgin and pregnant mice by real-time qRT-PCR (Figure 4D).

*Grb10* encodes an intracellular signalling molecule that can bind to and inhibit signalling from activated tyrosine-kinase receptors, notably the insulin receptor [144]. GRB10 is also regulated by and transduces signals downstream of mTORC1, a complex that couples extracellular signalling pathways to cell growth, metabolism and autophagy [145,146]. Genes encoding additional insulin signalling/mTORC1 pathway proteins were also significantly differentially expressed between the virgin and pregnant mice, all in a direction consistent with insulin resistance/reduced mTORC1 signalling ([147,148,149]; Figure 4E). As described above, *Grb10* expression is high during embryonic development of the pituitary but downregulated postnatally (Figure 1 and Figure 2). The analysis of the scRNAseq data indicated that there is low expression in the adult pituitary gland in somatotrophs and proliferating cells, as well as expression in the posterior pituitary, endothelial cells and connective tissue (Figure 4F). Immunohistochemistry on sections from the adult pituitary gland confirmed the low level of expression of GRB10 in hormone-producing cells, but strong staining was found in the posterior lobe in the cells adjacent to blood-filled spaces, consistent with an identity of endothelial cells (Figure 2Ac).

*Ddc* encodes the enzyme aromatic L-amino acid decarboxylase (AADC), which catalyses the decarboxylation of L-3,4-dihydroxyphenylalanine (L-DOPA) to dopamine, L-5-hydroxytryptophan to serotonin and L-tryptophan to tryptamine [150]. Since dopamine has an important role in the regulation of maternal PRL production during pregnancy and lactation [108], we examined the differentially expressed genes for other known regulators of PRL. We did not observe a significant difference in *Prl* gene expression between pregnant and virgin mice, as expected at this timepoint [109]. However, peptides that have previously been shown to regulate PRL secretion were observed to be modified by pregnancy (neuromedin U (NmU) [151] and vasoactive intestinal peptide (VIP) [152]; Figure 4G). The scRNAseq data indicated that *Ddc* expression is similar to that of *Grb10*, overlapping in the somatotroph and proliferating cell compartment (Figure 4H,I). Consistently, immunofluorescence revealed expression of DDC in a subset of GH-expressing somatotrophs, as well as in vascular cells of the pregnant gland (Figure 4J,K).

## 4. Discussion and Conclusions

### 4.1. Role of Imprinted Genes in Pituitary Development and the Imprinted Gene Network

Here we performed an unbiased study of imprinted gene expression in the developing and mature pituitary gland. We found that imprinted genes are amongst the highest expressed transcripts at both stages (Figure 1B), and that there is a subset of imprinted genes with very abundant embryonic transcription and limited or absent expression in the adult gland. Previous work has defined a subset of genes, the IGN, as co-ordinately downregulated in multiple organs, coincident with growth deceleration in early postnatal life [99]. We observed overlap between our “embryo high” genes and this growth deceleration IGN. We extended the temporal expression analysis by performing mRNA in situ hybridisation/immunohistochemistry and confirmed that *Igf2*, *Cdkn1c* and GRB10 are expressed in the developing RP in the SOX2^+^ cell population at stages when there is substantial gland proliferation and morphogenesis [81,82]. As the gland matures and cells commit to hormonal lineages, expression declines. At postnatal day 7 (P7) there was very low expression of all three genes (Figure 2), similar to previous IGN studies in other tissues [99], suggesting that the downregulation of imprinted genes in early postnatal life may be a feature of neuroendocrine organs as well as of tissues of mesendodermal origin. Mice with a paternal deletion of *Igf2* are growth restricted in utero and never reach normal weight, attaining ~60% wild-type body weight postnatally [153]. Maternal *Grb10* knock-out mice have increased growth during embryogenesis and retain their growth advantage into adulthood [154,155]. To our knowledge, the involvement of *Igf2/Grb10* on postnatal growth factor production by the pituitary axis has not been previously investigated. *Cdkn1c* encodes a cyclin-dependent kinase inhibitor that is known to limit G1/S progression [156,157]. Previous work demonstrated that the *Cdkn1c* dosage in the developing mouse pituitary gland determines the timing of the cell cycle exit of the SOX2^+^ progenitor population and the eventual size of the gland [64]. However, postnatal hormone production was not addressed. Other studies have shown that *Cdkn1c* expression dosage determines both pre- and postnatal weight in mice [158,159].

IGN genes could represent a transcriptional programme that confers potency to regenerate adult organs [107], an idea supported by the finding that imprinted genes are necessary for adult stem cell homeostasis in multiple contexts [102,103,104,105]. Utilising single-cell RNA sequencing data from adult mouse pituitaries [44], we asked if our embryo high genes were also expressed in the pituitary stem cell compartment. We observed no overlap between embryo high genes and genes that were defined as markers for the SOX2^+^ pituitary progenitor compartment. However, we did identify other IGN members and imprinted genes as stem cell markers. *Peg3* was enriched in the stem cell compartment, as were neighbouring genes in the same imprinting cluster, *Zim1* and *Usp29*. *Peg3* expression has been found in multiple stem/progenitor cell niches [106] and is necessary for regeneration of muscle, acting in the satellite cell compartment to regulate self-renewal [105]. Mice with a paternally inherited *Peg3* deletion are postnatally growth restricted [160,161], and females have defects in milk provision [160,162] (though this phenotype may be due to a locus-wide effect of the deletion [161]). Although *Peg3* expression is high in the hypothalamus and associated with oxytocin function [162], an action of *Peg3* on somatotroph/lactotroph homeostasis would also be consistent with the phenotypes of *Peg3* knock-out mice. Other stem cell markers included *Plagl1* and *Zdbf2*, both of which were also enriched in the proliferating cells (Appendix A). *Plagl1* encodes a zinc-finger transcription factor that binds to and regulates other imprinted and non-imprinted genes in the IGN, including *Cdkn1c*, *Igf2-H19* and *Grb10* [93,96]. *Plagl1* is expressed in the epithelial progenitor cells of the RP [163] and in the adult gland in stem- and hormone-producing cells [164]. *PLAGL1* negatively regulates cell proliferation and its loss of function has been reported in pituitary adenomas, where levels of PLAGL1 may mediate the response to somatostatin analogues that are commonly used to treat acromegaly (reviewed in [165]). The paternal deletion of *Plagl1* in mice causes embryonic growth retardation [94], but postnatal growth and pituitary hormone production in these mice has not been reported. *Zdbf2* is paternally expressed in the postnatal hypothalamus and pituitary of the mouse. Ablated *Zdbf2* function due to the deletion of a regulatory non-coding RNA causes postnatal growth retardation [166].

### 4.2. Pituitary IG Expression and Relevance to Human Disease

Altered expression of *GRB10* [13], *IGF2* [12] and *CDKN1C* [14] have all been implicated in SRS, a paediatric growth disorder that can manifest with postnatal growth retardation. We found that all three genes were highly expressed in the developing RP and their protein products could contribute to successful organogenesis. More work is required to establish if the misregulation of these “embryo high” genes has lifetime consequences for endocrine health.

PWS is a complex neuroendocrine disorder caused by the dysfunction of multiple paternally expressed genes on chromosome 15q11-13. PWS patients experience growth hormone deficiency (GHD) and postnatal growth retardation [31]. Evidence from murine models suggests that GHD is caused at least in part by pituitary hypoplasia—pituitary mass is reduced by ~40% in PWS imprinting centre deletion mice, with a concomitant reduction in GH content and circulating IGF1 [167]. The contribution of individual PWS cluster genes to the clinical phenotype has been intensively studied, employing detailed genetic analysis and murine models of individual gene deletions (recently reviewed in [168,169]). There is some evidence that loss of the *Magel2* function causes pituitary gland hypoplasia and GHD; patients with *MAGEL2* deletions can display congenital hypopituitarism [39] and *Magel2* knock-out mice are growth retarded with sex-specific defects in GH secretion. However, these mice also display considerable impairments to hypothalamic development that may impact on both the GH and the gonadotrophin axes [170,171,172]. We observed high *Magel2* expression during embryogenesis, but very low expression in adults, with no specific cell-type enrichment of gene expression (Figure 1D and Figure 3B). These data support a developmental role for *Magel2* in the pituitary gland. Similarly, *Mkrn3* had high embryonic expression in e13.5 RP, but was barely detectable postnatally (Figure 1D and Figure 3B). Many studies have described the expression of MKRN3 in the hypothalamic gonadotrophin system and its role in CPP (see above and Table 1); however, to date an animal model of *Mkrn3* deletion has not been described. Analysis of scRNAseq data from postnatal animals indicated that other PWS cluster genes might modulate the pituitary-gonadotrophin axis; *Snrpn* and *Ndn* were expressed abundantly in the postnatal AP and were significantly enriched in the gonadotroph lineage (Figure 3B,D, Table 2). Mice lacking a paternal copy of *Ndn* have impaired GnRH neuronal development [173,174], but pituitary gland involvement has not been explored in detail.

### 4.3. Imprinted Genes in the Pregnant Pituitary Gland

The expression dosage of imprinted genes as a class were not affected by pregnancy, at least at the late gestational timepoint that we examined (e15.5). Three imprinted genes were upregulated in the pituitary gland of the pregnant dam compared to the non-pregnant females: *Qpct*, *Grb10* and *Ddc*. *Qpct* encodes the glutamyl-peptide cyclotransferase, an enzyme that post-translationally modifies secretory neuropeptides such as TRH and GnRH [175]. *Qpct* is highly expressed in the pituitary gland [171], and the scRNAseq data showed significant enrichment in postnatal gonadotrophs (Figure 3D, Table 2). *Qpct* deletion in mice does not cause any major impairments to metabolism or fertility [176]. *Qpct* is imprinted in a tissue-specific manner, with maternal expression only in the placenta of all tissues assayed to date [140].

*Grb10* and *Ddc* are neighbouring imprinted genes on mouse chromosome 11 [177]. Both genes show complex allele-specific expression, with *Grb10* maternally expressed in most peripheral tissues and in the placenta, but paternally expressed in the CNS [142,143]. In humans, the direction and presence of imprinting in the brain and placenta appears conserved, but other tissues demonstrate biallelic expression [178]. To date, imprinted expression of *Ddc* has only been established in the embryonic heart, though there is evidence that additional embryonic tissues demonstrate a paternal allelic bias [141,177]. In our study of the pregnant pituitary gland, both genes were upregulated (Figure 4D). GRB10 negatively regulates signal transduction from tyrosine kinase receptors to downstream effectors such as mTORC1 [145,146]. We saw wider evidence of reduced insulin signalling/mTORC1 pathway activation in the pregnant pituitary gland (Figure 4E). Further experiments are required to understand any action of GRB10 in pituitary biology and maternal pregnancy adaptation. *Ddc* encodes aromatic L-amino acid decarboxylase (AADC), the final enzyme in the dopamine/serotonin/tryptamine biosynthesis cascade [150]. In mid-pregnancy, the PRL secretory pathway is completely suppressed by the action of placental lactogens that stimulate the hypothalamus to produce and transport dopamine to the pituitary, where it acts as a potent inhibitor [108]. Tyrosine hydroxylase is the rate-limiting enzyme in dopamine biosynthesis, and it is generally thought that this pathway is completed by dopaminergic neurons [108]. However, a recent study demonstrated that cells of the anterior pituitary gland can synthesise dopamine from L-DOPA by the local action of AADC. Moreover, locally produced dopamine was able to regulate PRL secretion [179]. We propose that our finding that *Ddc* is expressed in the adult pituitary gland and upregulated in pregnancy suggests an additional mechanism by which maternal PRL secretion is suppressed in gestation, and thus, merits further investigation.

### 4.4. Limitations and Conclusions

Our study has several limitations, the most important being that we did not establish the imprinting status/expressed allele of all the pituitary genes analysed. We compiled the list of known imprinted genes to maximise inclusivity without discarding genes with tissue-specific imprinting. This resulted in the inclusion of some genes that may only be imprinted in limited tissues (particularly the placenta) and may not share imprinting status between humans and mice. However, imprinting appeared to be maintained for many “canonically” imprinted genes in the human pituitary gland, based on the allele-specific RNAseq analysis of archived tissues publically-available datasets. Of interest, although *GRB10* is thought to be biallelically expressed in non-CNS tissues in humans [178], it appears to exhibit monoallelic pituitary expression [180]. Finally, the scRNAseq data from adult mice in this study was exclusively from males, limiting our ability to generalise, particularly with regard to gene expression localisation in the pregnant gland.

In conclusion, our review of endocrine phenotypes of IDs, together with the new data presented here of the temporal and spatial expression of imprinted genes, suggest that imprinting plays an important role in the development and function of the pituitary gland. Further studies of pituitary involvement in IDs may aid in understanding the underlying biology of these disorders and in suggesting additional therapeutic options. In addition, we presented evidence that imprinted gene dosage might co-ordinately regulate pituitary development and stem cell function, and that there may be roles for these genes, particularly in the gonadotrophin and prolactin pathways. Finally, we found dynamic expression of *Grb10* and *Ddc* in the pregnant gland. Further studies should be undertaken to determine if these genes are involved in maternal endocrine adaptations to pregnancy.

## Figures and Tables

**Figure 1 genes-12-00509-f001:**
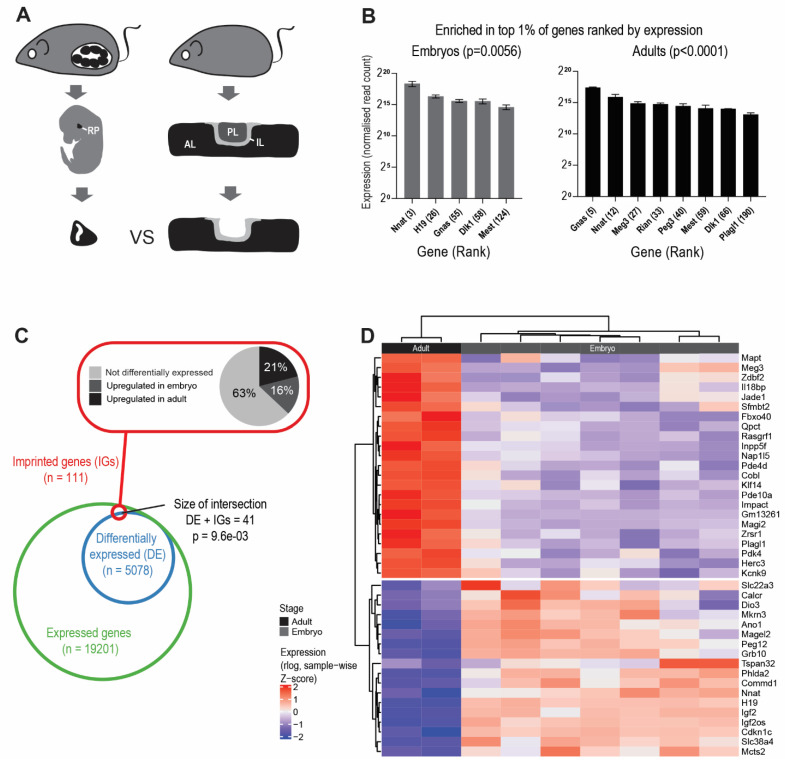
Imprinted genes are highly expressed in the anterior pituitary (AP) and are developmentally regulated. Schema of experimental design: for embryonic samples, Rathke’s pouch (RP) was microdissected from the embryonic heads at e13.5 (*n* = 7), adult samples at 8 weeks (*n* = 2) were dissected whole, then the posterior lobe (PL) was removed, leaving the anterior lobe (AL) and intermediate lobes (IL) for further analysis (**A**). We ranked genes according to mean normalised read count in the embryonic and adult pituitary samples. Imprinted genes at both ages were enriched in the top 1% of all expressed genes using Fisher’s exact test, and *p*-values are shown above the graphs (**B**). Venn diagram showing overlap of expressed imprinted genes (red) and differentially expressed genes (blue). The overlap was tested for statistical significance using Fisher’s exact test considering the list of expressed genes (green) as the background (**C**). Heatmap showing imprinted genes, which are differentially expressed between the embryo and adult samples (**D**). Statistical significance for differential expression between the embryo and adult samples was determined based on a *p*-value (corrected for multiple hypothesis testing based on the Benjamini–Hochberg procedure) threshold of 0.05 and a fold change threshold of 4. A regularised log transformation was applied to normalised counts (see Materials and Methods), and the values were then z-transformed across samples.

**Figure 2 genes-12-00509-f002:**
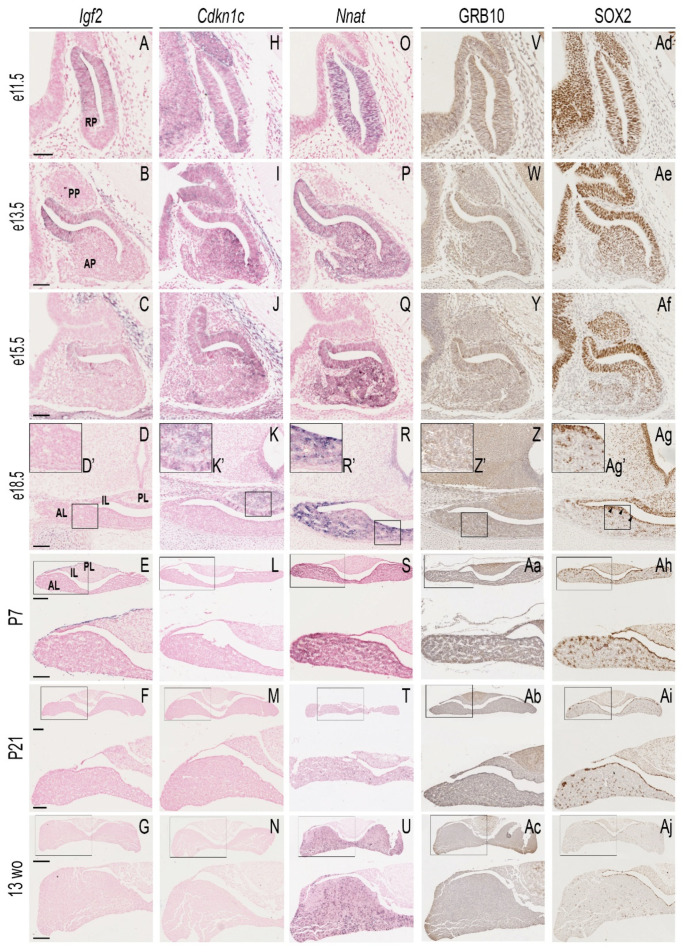
Expression of “embryo-high” imprinted genes in the developing and postnatal pituitary gland**.** In situ hybridisation (ISH) using antisense riboprobes against insulin growth factor 2 (*Igf2*) (**A**–**G**), cyclin-dependent kinase inhibitor 1C (*Cdkn1c*) (**H**–**N**), neuronatin (*Nnat*) (**O**–**U**) and immunohistochemistry (IHC) against growth-factor receptor bound protein 10 (GRB10) (**V–Ac**) and sex-determining region Y-related HMG box 2 (SOX2) (**Ad–Aj**) on the histological sections of wild-type pituitary glands collected at different developmental and postnatal stages. Samples at e11.5–e15.5 were cut sagittally, and embryos at e18.5 and postnatal samples were cut frontally. Postnatal pituitary glands were collected from female animals. **D’**, **K’**, **R’**, **Z’** and **Ag’** represent magnifications of the squared areas in D, K, R, Z and Ag, respectively. For the P7, P21 and 13-weeks-old (wo) rows, each picture contains a full image of the pituitary gland at the top and the magnified image of the boxed area at the bottom. *Igf2* showed expression around the cleft at e11.5 (**A**) and in the rostral tip of the anterior pituitary (AP) at e13.5 (arrowheads in **B**), but no expression was detected at later stages (**C**–**G**). Similarly, *Cdkn1c* showed expression during embryonic development (**H**–**K**), but no staining was observed for later stages (**L**–**N**). *Cdkn1c* staining was also observed in the infundibulum (Inf) (**H**). Staining was observed at e11.5 along the cleft and in the caudal area of the anterior pituitary at e13.5 and e15.5 (arrows in **I**–**J**, respectively), and in the posterior lobe (PL) and intermediate lobes (ILs) at e18.5 (**K** and **K’**). *Nnat* was widely expressed in Rathke’s pouch (RP) at e11.5 (**O**) and in the AP at all the stages analysed (**P**–**U**). GRB10 showed some faint staining in the AP (arrowheads in **Y**, **Z’**, **Aa**–**Ac**). Staining was also observed in the PL (arrows in Z, **Aa**–**Ac**). SOX2 was widely expressed in the RP and the Inf at e11.5. At e13.5, most cells in the AP and PL were SOX2^+^ (**Ae**). By e15.5, its expression started to become more restricted to the cells along the cleft (**Af**). By e18.5 and in the postnatal stages (**Ag**–**Aj**), SOX2 staining was clearly observed in the marginal zone (MZ) around the cleft, considered to be the stem cell niche of the pituitary gland. Abbreviations: AL, anterior lobe; AP, anterior pituitary; IL, intermediate lobe; Inf, infundibulum; MZ, marginal zone; PL, posterior lobe; RP, Rathke’s pouch. Scale bars in A, B and C represent 50 μm; scale bars in D represent 100 μm and 50 μm for the enlarged images; scale bars in E and F represent 200 μm (top) and 100 μm (bottom); scale bars in G represent 400 μm (top) and 200 μm (bottom) (*n* = 2).

**Figure 3 genes-12-00509-f003:**
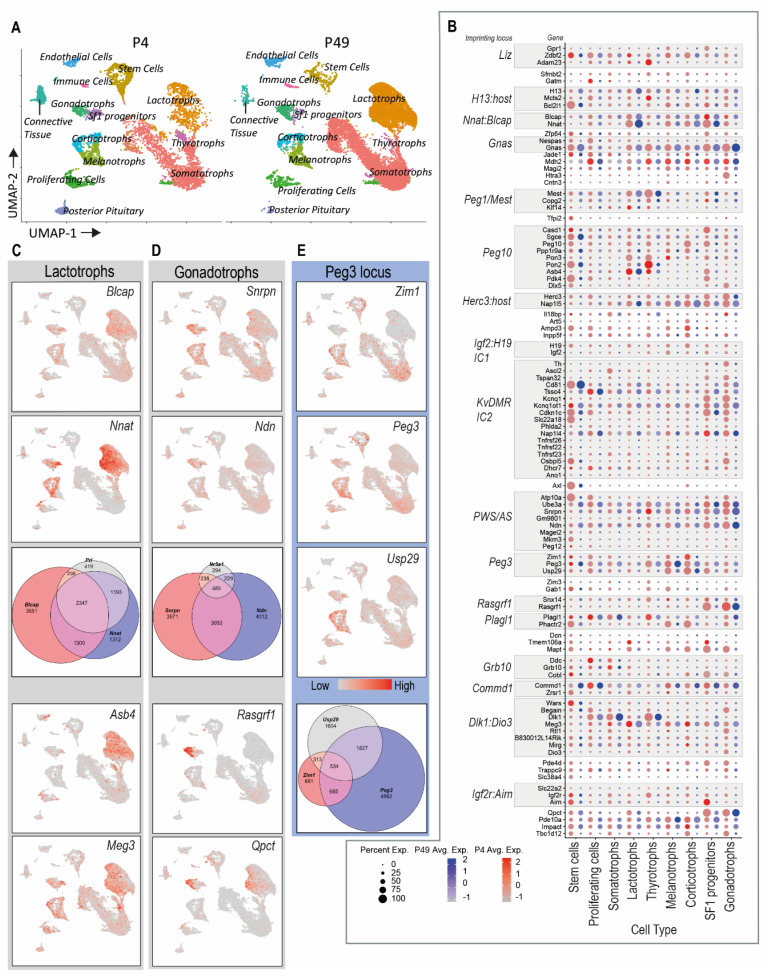
Imprinted gene expression in postnatal day 4 (P4) and postnatal day 49 (P49) anterior pituitary (AP) scRNAseq datasets. Single-cell RNA sequencing of postnatal day 4 (P4) and day 49 (P49) anterior pituitaries revealed 13 distinct cell types shared across the two ages (**A**). Then, 100 known imprinted genes (IGs) expressed in both ages were plotted in all identified cell types (**B**). IGs were grouped according to known IG cluster and relabelled if >1 member of the cluster was present in the dataset. Expression of IGs in P49 cells is shown in blue and P4 in red, whilst the size of the dot is indicative of the IG percentage expression in the specific cell type. Expression of the IGs significantly enriched in the lactotroph lineage (**C**). Linked bladder cancer-associated protein *(Blcap)* and *Nnat* showed overlapping expression in the prolactin (PRL)-expressing cells (Venn diagram). Enriched IGs in gonadotrophs. Prader–Willi syndrome (PWS) cluster genes small nuclear riboprotein polypeptide *n* (*Snrpn*) and necdin (*Ndn*) showed some overlap in cells expressing *Nr5a1* (gonadotrophs and steroidogenic factor 1 (SF1) progenitors, Appendix A) (**D**). Paternally expressed gene 3 (*Peg3*) and neighbouring genes zinc finger imprinted 1 (*Zim1*) and ubiquitin-specific peptidase 29 (*Usp29*) appeared to have overlapping expression, but at the cellular level this was limited (**E**). All three genes were significantly enriched in stem cells (Table 2). Venn diagrams show the circle size proportional to number of cells that express a target gene, and the number of cells is shown in the relevant circle. Grey to red indicates no expression to high expression.

**Figure 4 genes-12-00509-f004:**
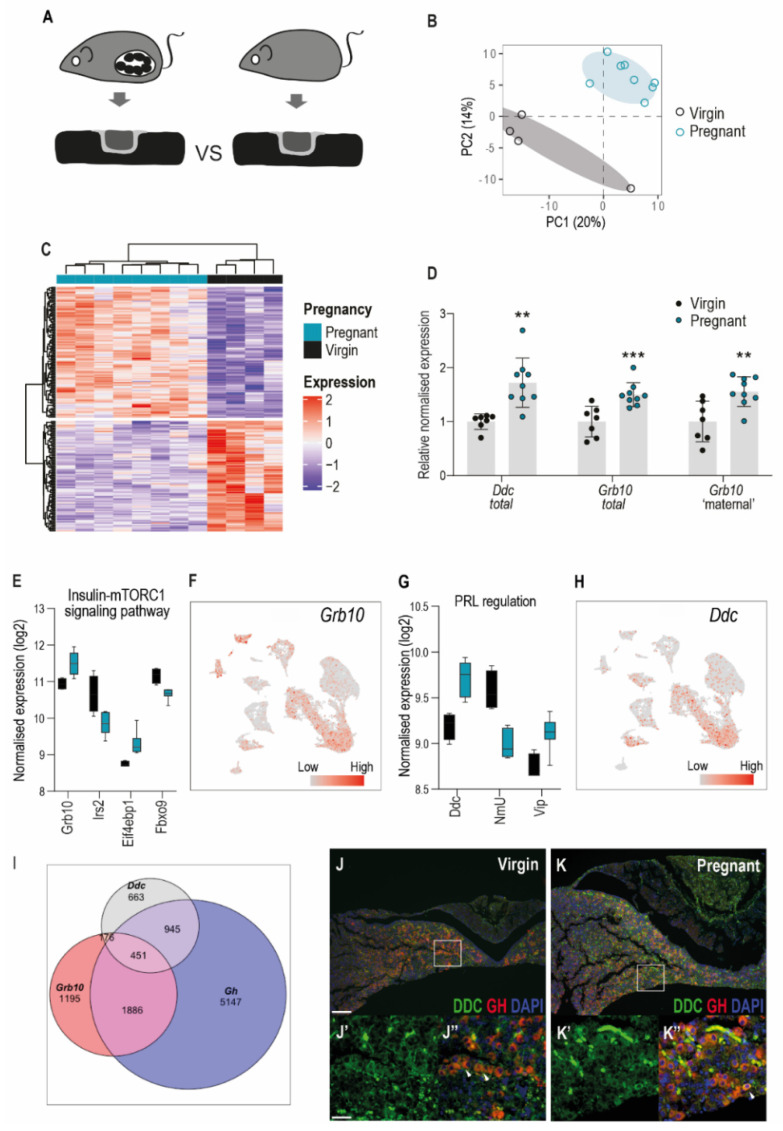
Transcriptional profiling of the pregnant pituitary gland at e15.5. Experimental schema: whole pituitary was removed from pregnant or virgin c57BL6/J mice at 3 months and compared (**A**). Non-supervised clustering indicates that the samples cluster according to experimental group (**B**). Heatmap showing genes which were differentially expressed in the whole pituitary of pregnant (*n* = 8) and virgin (*n* = 4) females (**C**). Statistical significance for differential expression between the groups was determined based on a *p*-value (corrected for multiple hypothesis testing based on the Benjamini–Hochberg procedure) threshold of 0.05 and a fold change threshold of 1.25. Probe signal intensities were quantile normalised and z-transformed across samples (see Materials and Methods). Real-time quantitative PCR validation of candidate gene expression in pituitaries from an independent cohort of e15.5 pregnant (*n* = 9) and virgin (*n* = 7) mice at 3 months (**D**). Data were normalised to housekeeping gene expression and depicted as mean relative expression ± SD (bars and error). Groups were called significantly different by Student’s t-test (** *p* < 0.01, *** *p* < 0.001). DE genes in the Insulin-mTORC1 pathway from the transcriptomics data shown in C (**E**). Feature plot showing the scRNAseq distribution of growth factor receptor-bound protein 10 (*Grb10*) (**F**). DE genes in prolactin (PRL) pathway from the transcriptomics data shown in C (**G**). Feature plot showing the scRNAseq distribution of *Ddc* (**H**). Venn diagram showing overlap of cells from scRNAseq data that express linked *Grb10* and *Ddc* and *Gh* (**I**). Immunofluorescence showing DDC expression in the adult virgin (**J**) and pregnant (**K**) gland with zoom into the anterior lobe parenchyma as single channel (**J’**,**K’**) and overlap of DDC, GH and DAPI signals (**J’’**,**K’’**). Overlap of DDC and GH immunoreactivity is evident in a subset of AP cells (white arrows). Box-and-whisker plots show the mean and min-max range of the data. Scale bars represent 500 μm for the low-magnification images and 50 μm for the enlarged images.

**Table 1 genes-12-00509-t001:** Summary of human imprinting disorders highlighting endocrine/pituitary phenotypes. Imprinted genes tend to cluster into distinct chromosomal loci, and the genetic and epigenetic lesions associated with these loci are linked to disease phenotypes. The loss or gain of function of the same subset of imprinted genes can have distinct phenotypes, such as 15q11-13 disruption, which may cause Prader–Willi syndrome (PWS) or Angelman syndrome (AS), depending on the parental origin of the lesion. See Table 2 for further information.

Location of Imprinting Cluster	Type of Mutation	Genes Affected	Normally Expressed Allele	Syndrome	Main Endocrine Features			Other Main Features	Ref
					Growth	Sexual Development	Metabolic Conditions		
6q24	UPD(6)pat/duplication of paternal allele/Hypomethylation of maternal DMR	*PLAGL1*	Paternal	Transient Neonatal Diabetes Mellitus [OMIM 601410]	Severe IUGR	Not reported	Hyperglycaemia, dehydration, absence of ketoacidosis		[6,7,8,9,10]
		*HYMAI*							
7p11.2-q13	UPD(7)mat	*GRB10*	Maternal	Silver–Russell [OMIM 180860]	IUGR, relative macrocephaly, postnatal growth failure	Premature adrenarche. Early and rapid puberty.	Perinatal feeding difficulties and hypoglycaemia. Develop insulin resistance.	Distinctive facial features (triangular shape, prominent forehead, narrow chin, small jaw). Clinodactyly.	[11,12,13,14,15,16,17]
11p15.5	Hypomethylation of IC1	*IGF-2*	Paternal						
		*H19*	Maternal						
	Loss-of-function mutations	*IGF-2*	Paternal						
	Gain of methylation IC2	*CDKN1C*	Maternal						
		*KCNQ1*	Maternal						
		*KCNQ1OT1*	Paternal						
11p15.5	Gain of methylation IC1	*IGF-2*	Paternal	Beckwith–Wiedemann [OMIM 130650]	Pre- and postnatal overgrowth.	Not reported	Neonatal hyperinsulinism	Macroglossia, abdominal wall defects. Predisposition to tumour development (Wilm’s tumour, adrenal carcinoma, hepatoblastoma) early in life. Visceromegaly. Renal abnormalities.	[18,19,20,21,22,23,24,25]
		*H19*	Maternal						
	Loss of methylation IC2	*CDKN1C*	Maternal						
		*KCNQ1*	Maternal						
		*KCNQ1OT1*	Paternal						
	Loss-of-function mutations	*CDKN1C*	Maternal						
11p15.5	Gain-of-function missense mutations	*CDKN1C*	Maternal	IMAGe [OMIM 614732]	IUGR	Genital abnormalities in males (micropenis, cryptorchidism, hypospadias).		Metaphyseal dysplasia. Adrenal insufficiency. Skeletal abnormalities	[26]
11p15.5	Missense mutations	*KCNQ1*	Maternal	Growth hormone deficiency (GHD) and gingival fibromatosis [OMIM 611010]	Small stature	Gonadotrophin deficiency		Gingival fibromatosis.	[27]
14q32	UPD14)mat	*DLK1* *RTL1* *DIO3* *MEG3* *MEG8*	Paternal Maternal	Temple [OMIM 616222]	IUGR. Postnatal short stature	Premature sexual development	Feeding difficulties in the neonatal period. Truncal obesity	Muscular hypotonia, motor and mental developmental delay, scoliosis	[28]
	UPD14)pat			Kagami–Ogata [OMIM 608149]	Postnatal growth retardation				[29]
14q32	Inactivating mutations, deletions	*DLK1*	Paternal	Central precocious puberty		Premature sexual development	Truncal overweight/obesity, insulin resistance, T2DM, hyperlipidaemia		[30]
15q11-q13	Deletion of paternal region/mUPD	*MKRN3* *MAGEL2* *NDN* *NPAP1* *SNRPN* *SNORD116*	Paternal	Prader–Willi [OMIM 176270]	Short stature. Reduced GH. Reduced IGF-I Low bone density.	Variable hypogonadism phenotype (genital hypoplasia, incomplete pubertal development, infertility) Low to normal levels of testosterone or oestrogen, FSH and LH. GnRH insensitivity	Hyperphagia	Mild intellectual disability, obsessive-compulsive traits	[31,32,33,34,35]
15q11-q13	Deletion of maternal region/UPD(15)pat	*UBE3A*	Maternal	Angelman [OMIM 105830]				Intellectual disability, Speech impairment, gait ataxia	[36,37]
15q11-q13	Point mutations, deletions, frameshifts	*MAGEL2*		Schaaf–Yang [OMIM 615547]	GHD. Short stature.	Gonadotrophin deficiency	Hyperinsulinaemic hypoglycaemia. Neonatal feeding difficulties, followed by hyperphagia. Central hypothyroidism	Panhypopituitarism associated with a hypoplastic anterior pituitary gland. Adrenal insufficiency. Arthrogryposis. ASD. Intellectual disability	[38,39]
15q11-q13	Inactivating mutations	*MKRN3*	Paternal	Central precocious puberty		Premature reactivation of the reproductive axis.			[40]
20q13.2-13.3	Activating mutations	*GNAS*		McCune–Albright [OMIM 174800]	Acromegaly (caused by GH-secreting pituitary tumours)	Sexual precocity	Hyperthyroidism (caused by hyperactive thyroid nodules) Hypercortisolism (associated with macronodular adrenal hyperplasia or adrenal adenomas)	Hyperpigmentation of the skin Osteomalacia	[41,42]

## Data Availability

Data available in a publicly accessible repository The data presented in this study are openly available in the Gene Expression Omnibus (GEO) at (https://www.ncbi.nlm.nih.gov/geo/query/acc.cgi?acc=GSE1169597; https://www.ncbi.nlm.nih.gov/geo/query/acc.cgi?acc=GSE171060), reference number (GSE169597; GSE171060).

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
