# Peer review of "Dynamic Expression of Imprinted Genes in the Developing and Postnatal Pituitary Gland"

_genes, 2021, doi:10.3390/genes12040509_

Round 1
Reviewer 1 Report
This manuscript by Scaglotti and colleagues addresses a gap in the field of imprinting and endocrine function/disorders. Here the investigators review existing data on imprinted genes involved in endocrine function. They also describe imprinting disorders that clearly involve endocrine organs, i.e. those presenting with a variety of growth, developmental and gonads disorders. They then integrate this reviewed information with gene expression data generated by themselves from the pituitary gland and re-analysis of single cell data from others. Interestingly, imprinted genes are among the most highly expressed genes in the pituitary. This works shows that imprinted genes have central and critical roles in endocrine function. In addition to providing a unique review of the literature of imprinted genes and the endocrine system, the authors have generated and reanalyzed valuable (new) data that will be of use to the research community.
I only have a few questions/comments:
For Table 1, please use “intellectual disability” in place of “mental retardation.”
Table 1, Chr14 syndromes are “UPD” not “UDP”
Approximately Line 267: In the discussion of gonad disorders for PWS, do all PWS individuals exhibit such disorders? Some PWS have small deletions in the snoRNA genes.
Line 520- and Figure 4, S5: It is stated that the imprinted forms of Grb10 and Ddc overlap and are induced in the pregnant dam in the pituitary. Whereas the Grb10 somatic form is shown specifically in figure 4, it is unclear to me what form of Ddc is shown. Is that total Ddc?
Author Response
#Reviewer 1.
This manuscript by Scaglotti and colleagues addresses a gap in the field of imprinting and endocrine function/disorders. Here the investigators review existing data on imprinted genes involved in endocrine function. They also describe imprinting disorders that clearly involve endocrine organs, i.e. those presenting with a variety of growth, developmental and gonads disorders. They then integrate this reviewed information with gene expression data generated by themselves from the pituitary gland and re-analysis of single cell data from others. Interestingly, imprinted genes are among the most highly expressed genes in the pituitary. This works shows that imprinted genes have central and critical roles in endocrine function. In addition to providing a unique review of the literature of imprinted genes and the endocrine system, the authors have generated and reanalyzed valuable (new) data that will be of use to the research community.
Our Response:
We are delighted to hear that Reviewer 1 sees our manuscript as a valuable contribution to the field of imprinting genes and their roles in endocrine function and we thank Reviewer 1 for their comment.
#Reviewer1: I only have a few questions/comments:
- For Table 1, please use “intellectual disability” in place of “mental retardation.”
We have changed the wording to ‘intellectual disability’ (Table 1, column 9, p.24) - Table 1, Chr14 syndromes are “UPD” not “UDP”
We thank you the reviewer for noticing the misspelling. We have changed it to the correct ‘UPD’ for Chr14 syndromes, as well as for 6q24, and 15q11-q13.
- Approximately Line 267: In the discussion of gonad disorders for PWS, do all PWS individuals exhibit such disorders? Some PWS have small deletions in the snoRNA genes.
We have expanded some of the information regarding PWS in Table 1. Specifically, we have added SNORD116 to the list of possible deletions occurring in the 15q11-q13 region (Sahoo et al. Nat Genet. 2008; de Smith et al. Hum Mol Genet. 2009). We have also modified the section of sexual development for PWS (Table 1). Although all PWS patients present some degree of hypogonadism, we agree that the range of phenotypes is quite broad. To reflect this, we have now changed the description to “Variable hypogonadism phenotype”. We have also changed to “low to normal levels of testosterone or estrogen, FSH and LH”, in agreement with clinical findings showing the high range of variability in the sexual hormone levels in PWS (Brandau, et al. Am J Med Genet. 2008; reviewed in Heksch et al, Transl Pediatrics, 2017).
- Line 520- and Figure 4, S5: It is stated that the imprinted forms of Grb10 and Ddc overlap and are induced in the pregnant dam in the pituitary. Whereas the Grb10 somatic form is shown specifically in figure 4, it is unclear to me what form of Ddc is shown. Is that total Ddc?
We agree with the reviewer that this could generate confusion. We have now changed the label in Fig.4D to ‘Ddc total’, and we have also specified that the pair of primers used for the analysis is designed to detect Ddc (total) in Table S3.
Reviewer 2 Report
In the present study entitled “Dynamic Expression of Imprinted Genes in the Developing and Postnatal Pituitary Gland” Scagliotti et al. aimed at describing the imprinted transcriptome of the anterior pituitary gland by the means of the analysis of RNA-seq data. The authors analyzed novel transcriptomic data from the embryonic, adult, and pregnant pituitary gland to fully describe the changes in expression of imprinted genes in the murine pituitary gland both during development and pregnancy. In addition, the authors reanalyzed previously published transcriptomic datasets to validate their observation and give a more comprehensive description of the expression of imprinted genes in the murine pituitary gland.This work, which is rather original and technically well-conducted, gives an interesting view of imprinted gene networks in the pituitary gland, in different life phases. The experimental workflow, however, is not always completely clear. In my opinion, the Introduction section is too long and there is a certain grade of redundancy in the Results and Discussion section [e.g. the function of some genes, such as Grb10, has been described at least twice (in the paragraphs starting at lines 524 and 670]. A more comprehensive description of some of the choices that guide the authors along this work would have been instead desirable.
Below, a list of comments and/or points/aspects for the authors to review:
- Comparing embryo vs adult pituitary, the authors analyzed this latter as a whole. Is there any possibility that a compensating effect may hide some differentially expressed genes? In other words, whether a gene were highly expressed only in a single cell population (e.g. somatotrophs) is it possible it would not be identified by this approach?
- To test their hypothesis that some “embryo high gene”s could be associated with the proliferating progenitor population the authors evaluated Igf2, Cdkn1c, and Nnat using one approach (e.g. ISH), while GRB10 and SOX2 with another one (e.g. IHC). Which are the reasons for such a choice? Can they exclude the introduction of a bias comparing two different molecular species?
- The authors claimed that imprinted genes are often co-regulated within genomic clusters. How did they explain the lack of such coordinated expression, except for the Igf-H19 cluster, in the developing pituitary? Moreover, comparing embryo vs adult pituitaries, the authors identified 18 and 24 out of 111 imprinted genes with higher expression in either the former or the latter, respectively. They then further evaluated five of the embryo high genes. The reasons that led to this choice are, however, not completely clear.
- The authors generate expression data to compare IG in Embryo vs adult and Pregnant vs virgin. Which is the reason they choose two different experimental approaches – i.e. RNAseq and microarray analysis – to generate these data? Why did they limit their validation of DE targets only to microarray emerging data?
Minor
In the methods section, the authors reported that “8 week-old c57BL6/J females from 4 independent litters were randomly allocated to mating or virgin groups”, while in Figure 4c expression data from eight pregnant and four virgins are reported.
Author Response
#Reviewer 2.
In the present study entitled “Dynamic Expression of Imprinted Genes in the Developing and Postnatal Pituitary Gland” Scagliotti et al. aimed at describing the imprinted transcriptome of the anterior pituitary gland by the means of the analysis of RNA-seq data. The authors analyzed novel transcriptomic data from the embryonic, adult, and pregnant pituitary gland to fully describe the changes in expression of imprinted genes in the murine pituitary gland both during development and pregnancy. In addition, the authors reanalyzed previously published transcriptomic datasets to validate their observation and give a more comprehensive description of the expression of imprinted genes in the murine pituitary gland.This work, which is rather original and technically well-conducted, gives an interesting view of imprinted gene networks in the pituitary gland, in different life phases. The experimental workflow, however, is not always completely clear.
Our response:
We thank the reviewer for their comment. We are pleased that our work has been defined as “rather original and technically well-conducted”.
In my opinion, the Introduction section is too long and there is a certain grade of redundancy in the Results and Discussion section [e.g. the function of some genes, such as Grb10, has been described at least twice (in the paragraphs starting at lines 524 and 670]. A more comprehensive description of some of the choices that guide the authors along this work would have been instead desirable.
We acknowledge that this manuscript contains an extended Introduction section. Indeed, the idea behind this work was to produce a “hybrid manuscript” that would combine:
- an extensive critical review of the published literature to highlight the roles of imprinting genes in endocrine systems, since this has not been done before;
- some original work done by using previously published and unpublished data available to our group, to fill some of the gaps in the field of imprinting genes and their role in pituitary development and function.
For this reason, we have chosen to dedicate more space to the Introduction/review part than might be ordinarily found in a data paper. However, we agree with the reviewer that there were some repetitions. We have deleted “a complex that promotes cell growth, metabolism and autophagy” from line 563, as this was already stated in line 442.
We acknowledge that the experimental workflow and the reasons behind our choices could have been clearer. We hope that the changes made to the manuscript and the answer to the points raised below have now improved the clarity of our experimental workflow.
Below, a list of comments and/or points/aspects for the authors to review:
- Comparing embryo vs adult pituitary, the authors analyzed this latter as a whole. Is there any possibility that a compensating effect may hide some differentially expressed genes? In other words, whether a gene were highly expressed only in a single cell population (e.g. somatotrophs) is it possible it would not be identified by this approach?
We agree with the reviewer that it is indeed possible that a compensating effect could affect the results of this analysis. Although we have looked at the imprinted genes that were expressed in the different subpopulations of cells in the single-cell datasets at P4 and P49, we have not compared each sub-population with the embryonic bulk RNAseq data. We think that future and more detailed analyses are needed and will highlight more aspects of the role of imprinted genes in the pituitary.
- To test their hypothesis that some “embryo high gene”s could be associated with the proliferating progenitor population the authors evaluated Igf2, Cdkn1c, and Nnat using one approach (e.g. ISH), while GRB10 and SOX2 with another one (e.g. IHC). Which are the reasons for such a choice? Can they exclude the introduction of a bias comparing two different molecular species?
We agree with the reviewer that the use of one unique technique would have been ideal and that we cannot exclude the introduction of a bias. However, we think that IHC is more informative when looking at gene expression, since it detects the final molecular product of the gene (protein) and provides extra information about the cell localisation of the target protein. Therefore, where good antibodies were available (GRB10-SOX2), we performed IHC. For all the other targets (Nnat, Igf2 and Cdkn1c), we used ISH. To explain our reasoning to future readers, we have added the following phrase in the Method section (line 644): “Expression analyses on histological sections were carried out using ISH when working antibodies for the selected targets were not available.”
- The authors claimed that imprinted genes are often co-regulated within genomic clusters. How did they explain the lack of such coordinated expression, except for the Igf-H19 cluster, in the developing pituitary? Moreover, comparing embryo vs adult pituitaries, the authors identified 18 and 24 out of 111 imprinted genes with higher expression in either the former or the latter, respectively. They then further evaluated five of the embryo high genes. The reasons that led to this choice are, however, not completely clear.
Although imprinted genes are indeed often co-regulated within genomic clusters, this is not always the case and it generally depends on the genes sharing the same enhancers in that specific setting. For example, the co-expression of Igf2-H19 in endodermal tissues is well described and the common enhancers have been mapped (Leighton, 1995 PMID: 7544754). In other clusters the extent of co-expression is less clear, for example at the Dlk1-Meg3 cluster there is clear co-expression of the genes in some embryonic tissues (e.g somites and developing muscle), in other tissues there is no overlap (e.g. liver), daRocha 2007 PMID: 17449025. Understanding the extent to which imprinted genes share enhancer sequences within clusters, and how this relates to the imprinting mechanism awaits further work utilising high resolution chromatin-confimation-capture technologies. However, our work and others invites caution about the assumption of co-regulation in all tissues when designing such studies.
Regarding the choice of imprinted genes further evaluated, Cdkn1c, Grb10 and Nnat were amongst the top 4 most up-regulated genes identified in our analysis. We then chose to look at Igf2 (also in the list of the most upregulated genes), as it is a member of the Imprinted Gene Network which includes Cdkn1c, Grb10 and Nnat (Varrault, Dev Cell 2006). Unfortunately, for technical reasons we were not able to further explore H19 expression in this work, although we recognise its importance.
- The authors generate expression data to compare IG in Embryo vs adult and Pregnant vs virgin. Which is the reason they choose two different experimental approaches – i.e. RNAseq and microarray analysis – to generate these data? Why did they limit their validation of DE targets only to microarray emerging data?
The microarray dataset was generated in our lab several years ago (but never published), when the RNAseq technology was neither so commonly used nor so affordable. All the other datasets have been generated more recently, and for this reason RNAseq technology was used.
Regarding the validation of DE targets, there is a technical limitation due to the anatomical size of the pituitary gland at E13.5 and the consequent amount of RNA that can be extracted from it, which would make the validation of several genes of interest challenging. Therefore, to validate the DE targets identified by comparing embryo vs adult pituitaries, we have opted for the expression analysis using IHC and ISH on tissue sections, which has also allowed us to study how the expression evolved throughout pituitary development.
Minor
In the methods section, the authors reported that “8 week-old c57BL6/J females from 4 independent litters were randomly allocated to mating or virgin groups”, while in Figure 4c expression data from eight pregnant and four virgins are reported.
We agree that this paragraph could have been clearer. The ‘8’ at the beginning of the phrase was actually referring to the age of the mice. We have now modified the phrase to “8-week-old c57BL6/J females from 4 independent litters were randomly allocated to mating (n=8) or virgin (n=4) groups” (line 611). We have also specified the number of animals per experimental group in the legend of Fig.4 (line 423).